# CodonMPNN for Organism Specific and Codon Optimal Inverse Folding

**Hannes Stark** [* 1]   **Umesh Padia** [* 1]   **Julia Balla** [1]   **Cameron Diao** [1]

## Abstract

Generating protein sequences conditioned on protein structures is an impactful technique for protein engineering. When synthesizing engineered proteins, they are commonly translated into DNA and expressed in an organism such as yeast. One difficulty in this process is that the expression rates can be low due to suboptimal codon sequences for expressing a protein in a host organism. We propose CodonMPNN, which generates a codon sequence conditioned on a protein backbone structure and an organism label. If naturally occurring DNA sequences are close to codon optimality, CodonMPNN could learn to generate codon sequences with higher expression yields than heuristic codon choices for generated amino acid sequences. Experiments show that CodonMPNN retains the performance of previous inverse folding approaches and recovers wild-type codons more frequently than baselines. Furthermore, CodonMPNN has a higher likelihood of generating high-fitness codon sequences than low-fitness codon sequences for the same protein sequence. Code is available at https://github.com/HannesStark/CodonMPNN.

## 1. Introduction

A significant barrier for protein engineering and protein production is the expression of engineered RNA in host systems such as yeast (Presnyak et al., 2015). One aspect leading toward low expression yields is the suboptimality of the DNA/codon sequence used to express the protein: as illustrated in Figure 1, multiple codon sequences can encode the same protein sequence, but each codon sequence can interact differently with its environment in a cell. This can lead to different behaviors between codon sequences that encode the same protein, such as different expression levels.

---

[*]Equal contribution [1]CSAIL, Massachusetts Institute of Technology, Cambridge, MA, USA. Correspondence to: Hannes Stark <hstark@mit.edu>.

*Accepted at the 1st Machine Learning for Life and Material Sciences Workshop at ICML 2024. Copyright 2024 by the author(s).*

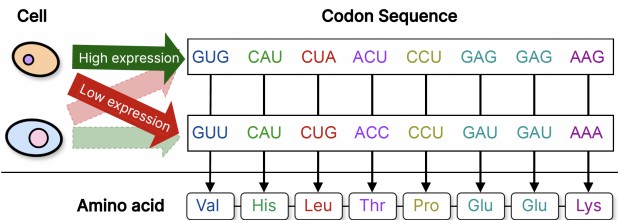

*Figure 1.* Amino acid sequences have corresponding DNA sequences with triplets of nucleotides (A, C, G, U) corresponding to amino acids. Since there are 64 possible triplets called codons and only 20 amino acids, there are multiple codon sequences for each protein sequence. Some have higher expression rates than others, and some are not expressed at all. This expression level depends on the host in which the codon sequence is expressed.

This suboptimality of codon sequences is also a bottleneck in the recently prominent protein engineering approach of generating protein sequences conditioned on a protein backbone structure with tools such as ProteinMPNN (Dauparas et al., 2022) to improve stability, function, or to use in denovo protein design (Watson et al., 2023). When validating the proteins obtained from such tools by synthesizing, expressing, and evaluating them, the obtained amino acid sequences also need to be translated into codon sequences for which optimization methods exist (Bahiri-Elitzur & Tuller, 2021). However, these optimization methods are imperfect and often fail for less well-studied hosts.

Furthermore, the following workflow misses an opportunity for optimization toward higher expression yields: 1) obtain protein structure with hypothesized desired function, 2) generate amino acid sequence that folds into the structure, 3) obtain codon sequence that is optimal for the amino acid sequence. This workflow does not allow for changing the amino acid sequence such that the codon sequence encoding the structure has higher expression yields.

To address these shortcomings, we propose CodonMPNN, which generates codon sequences conditioned on a protein backbone structure and an organism label for the host system. Assuming that *naturally occurring DNA sequences are closer to codon optimality than a random codon sequence that encodes the same structure*, CodonMPNN trained on natural protein codon sequences will generate

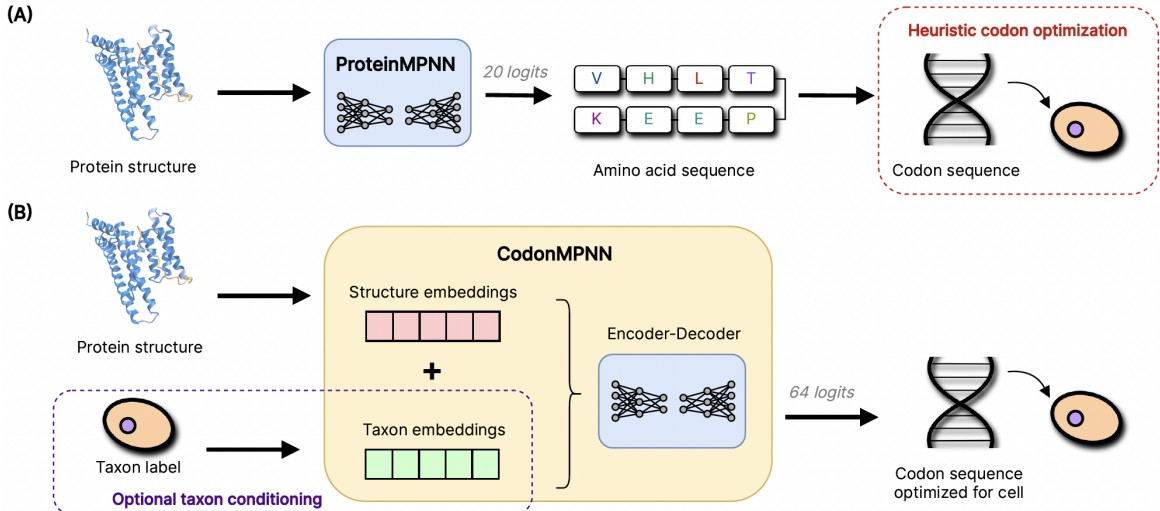

*Figure 2.* **CodonMPNN overview.** In the prevailing approach **(A)**, an inverse folding model, such as ProteinMPNN, generates an amino acid sequence. For experimental validation, this is mapped to a codon sequence (DNA sequence) via heuristic codon optimization tools and expressed in a specific system. As an alternative, we propose CodonMPNN **(B)**, which directly generates codon sequences conditioned on a structure and the taxon label of the host organism in which the codon sequence should be expressed.

high-expression codon sequences.

Empirically, CodonMPNN trained on the AlphaFold database produces recovery rates and designability evaluation metrics similar to ProteinMPNN (Dauparas et al., 2022). Further, the codon recovery rate of CodonMPNN is higher than that of sequences generated by translating our generated codon sequences to amino acids and choosing the most frequent codon for that amino acid. Lastly, we show that CodonMPNN has higher likelihoods for codon sequences with high fitness than for codon sequences with lower fitness, even if the sequences encode the same protein.

## 2. Related Work

*Inverse folding*, with a different meaning in the biology community, has been adapted to describe generating a protein sequence conditioned on a protein structure. For this task, several generative models with different factorizations of the sequence distribution and architectures have been proposed (Dauparas et al., 2022; Hsu et al., 2022; Shanker et al., 2023; Gao et al., 2023) and found considerable success in protein engineering (Watson et al., 2023; Sumida et al., 2024; Wang et al., 2024). These current approaches do not use the information about the host system in which the engineered protein will be expressed.

Previous work on generative models for codon sequences includes codon language models (Outeiral & Deane, 2024), which replace the vocabulary of protein language models with codons. Furthermore, Yang et al. (2019) conducted preliminary explorations of generating codon sequences conditioned on an amino acid sequence.

## 3. Method

CodonMPNN uses the same framework as ProteinMPNN (Dauparas et al., 2022), adapted to predict 64 codons instead of 20 residue types, and with the additional option of conditioning on the host system in which the DNA should be expressed. Thus, CodonMPNN is an any-order autoregressive model (Shih et al., 2022) over codon sequences conditioned on protein structures. To state this in more detail, let $s \in \{1, \ldots, 64\}^L$ be a codon sequence and $x \in \mathbb{R}^{L \times 4 \times 3}$ a protein structure of 3D coordinates of $L$ residues with their 4 backbone atoms. We train Codon-MPNN to predict $p(s_{\sigma(i)} \mid s_{\sigma(<i)}, x; \sigma)$ for any permutation $\sigma : \{1, \ldots, L\} \mapsto \{1, \ldots, L\}$ and $i \in \{1, \ldots, 64\}$. From this, we construct and sample an autoregressive factorization of the density over codon sequences $p(s) = \prod_{i=1}^{L} p(s_{\sigma(i)} \mid s_{\sigma(<i)}, x; \sigma)$ for any order $\sigma$.

**Architecture.** We use ProteinMPNN's encoder-decoder architecture where the encoder takes the protein structure $x$ as input and produces structure embeddings which the decoder uses together with the partial codon sequence $s_{\sigma(<i)}$ to predict $p(s_{\sigma(i)} \mid s_{\sigma(<i)}, x; \sigma)$. Both the encoder and decoder are 3 message passing neural network layers (Gilmer et al., 2017) over a graph with residues as nodes. Each node has edges to the 48 nearest nodes in terms of Euclidean distance between alpha carbons. Next to the graph connectivity, the only information drawn from the protein structure are edge features constructed from all pairwise distances between the backbone atoms of connected residues. Thus, the predicted probabilities are invariant with respect to Euclidean transformations of $x$. We implement the decoder's autoregressive prediction by masking messages from residues that have

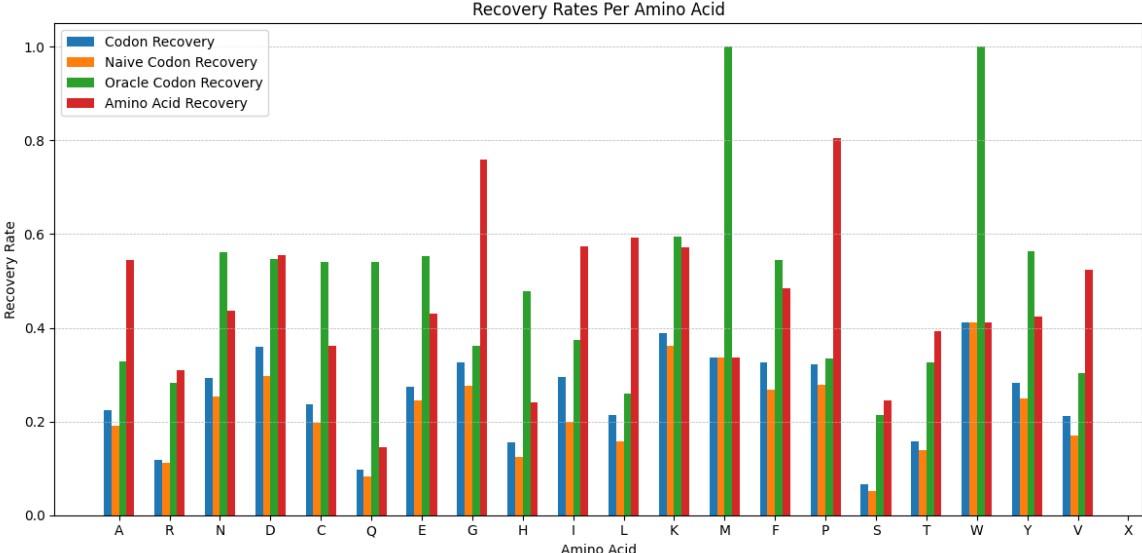

*Figure 3.* **Recovery rates per amino acid types.** *Codon Recovery* and *Amino Acid Recovery* show the recovery rates of CodonMPNN's generated sequences. *Naive Codon Recovery* is the recovery rate of codon sequences obtained by translating CodonMPNN's codons to amino acids and choosing their most frequent codons. *Oracle Codon Recovery* shows the same for the ground truth amino acids.

a higher index in the permutation $\sigma$ than the residue that receives the messages.

**Taxon Conditioning.** The diversity of cellular environments among organisms leads to different expression levels of a codon sequence depending on the host. Typically, the host cell for protein expression is *predetermined and known* to the user of the inverse folding model, providing an opportunity to condition generated sequences to suit the particular cellular environment. As a mapping from protein to host organism, we use the taxonomy map of the National Center for Biotechnology Information (NCBI) (Schoch et al., 2020). This database provides a hierarchical classification of organisms, enabling us to group organisms into clusters with common cellular environments.

A naive approach to clustering the tree is to group all organisms by shared ancestors corresponding to a specified taxonomic rank (e.g., phylum or order). However, for any given rank, this leads to imbalanced cluster sizes and missing assignments for organisms without a corresponding ancestor. Very large clusters would group cellular environments with different expression behaviors together. Furthermore, many proteins (e.g., 6,826 for the choice of order as taxonomic rank) would fall into clusters of size 1 with overly specific expression environments that will not generalize. Lastly, under this naive partitioning, the taxon identifier of a host system at test time might fall into its own cluster that was never observed during training, and CodonMPNN would be unable to learn which codon sequences are preferential for the targeted cellular environment.

To obtain more useful taxon clusters to condition Codon-

MPNN on, we instead implement a tree partitioning algorithm that creates approximately balanced subgroups while still preserving hierarchical information. Given a tree on $n$ nodes and a desired number of clusters $k$, the algorithm traverses the tree and tries to keep all subtrees in the same cluster without exceeding the ideal cluster size $\lceil \frac{n}{k} \rceil$. The full algorithm is described in appendix A. After obtaining a $k$-clustering of the taxons, we pass the cluster labels into an embedding layer and add the output to the initial node and edge embeddings of CodonMPNN. Additionally, for 50% of the training examples, we pass a null token to Codon-MPNN that indicates the absence of a taxon label and can be employed for generating a sequence from the marginal distribution without taxon conditioning.

## 4. Experiments

**Data:** We train and evaluate CodonMPNN (and Protein-MPNN for comparison) on AFDB (Varadi et al., 2024) structures with pLDDT> 0.9. We cluster the proteins to 30% sequence identity and choose one representative for each cluster. The test set has a maximum of 30% sequence identity to the training data.

**Metrics:** We evaluate recovery rates, designablity TM-Scores, and model likelihoods. Recovery rates are the percentage of generated codons or amino acids that match the wild-type sequence of a structure. Furthermore, we report the TMScore between the input structure and the structure that ESMFold predicts for a generated sequence. Assuming ESMFold's correctness, this evaluates whether the generated sequence takes on the same structure as the model input

*Table 1.* **CodonMPNN recovery rates and designability.** Shown are the fraction of recovered codons as *Codon %*, and amino acids as *AA %*, and the TM-Score as *TM* between the input structure and ESMFold's predicted structure for the models' generated sequence.

|                      | CODON % | AA %  | TM   |
|----------------------|---------|-------|------|
| PROTEINMPNN          | 20.5%   | 49.8% | 0.83 |
| PROTEINMPNN-TAXON    | 20.8%   | 50.3  | 0.86 |
| CODONMPNN            | 24.8%   | 49.4% | 0.84 |

structure and is a widely used indicator of "designability" (Yang et al., 2023; Campbell et al., 2024).

**Question 1: Does CodonMPNN retain ProteinMPNN's performance?** Table 1 shows that CodonMPNN recovers wild-type amino acids of a protein structure as frequently as ProteinMPNN and has the same designability in terms of TM-Score between the input structure and refolded structure. We additionally train ProteinMPNN with the same taxon conditioning as CodonMPNN (ProteinMPNN-taxon in the table) to verify that CodonMPNN also retains any potential performance improvement in ProteinMPNN with taxon conditioning and find the metrics to largely stay the same. Thus, CodonMPNN retains ProteinMPNN's performance.

**Question 2: Does CodonMPNN improve over choosing the most frequent codon per amino acid?** The codon recovery rates in Table 1 show that CodonMPNN more frequently recovers the wild-type codon than determining the codon sequence via choosing the most frequent codon per amino acid that was generated from ProteinMPNN. Moreover, the codon recovery obtained from CodonMPNN by translating it to an amino acid sequence and then choosing the most frequent codon for each amino acid is 20.9%. This is distinctly lower than CodonMPNN's 24.8% recovery rate. Similar insights can also be attained per amino acid from Figure 3, which shows for which amino acids CodonMPNN's choice of codon differs the most from the naive frequency-based codon choice.

**Question 3: Does CodonMPNN have higher likelihoods for highly expressed codon sequences than for low expression codon sequences that encode the same protein?**

To answer this question, we use yeast mutant data from (Shen et al., 2022). This data contains the fitness effects of 8,341 point mutations in 21 genes in budding yeast. In particular, the authors found that synonymous mutations, which alter the codon sequence but not the resulting protein sequence, can still have significant fitness effects. We only consider the 250 synonymous mutations with the most significant fitness effects, measured as the absolute difference between wild type and mutant fitness levels.

For each wild type sequence, we predict the corresponding protein structure using Alphafold 2 (Jumper et al., 2021;

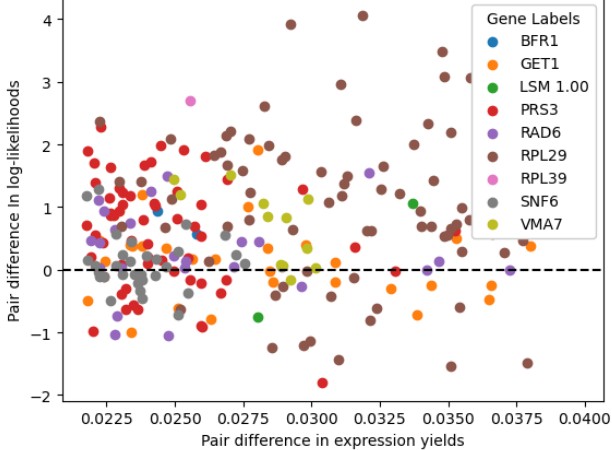

*Figure 4.* **Likelihoods For Synonymous Coding Sequences.** Each point is a pair of synonymous coding sequences. Points above the dashed line correspond to correct predictions. *Pair difference in expression yields* is the difference between the higher and lower expression yields of sequences in each pair. *Pair difference in log-likelihoods* is the difference in log-likelihoods between the highly- and lowly-expressed sequences.

Ahdritz et al., 2022). This structure is passed as conditioning information to CodonMPNN, which then predicts likelihoods for both the wild-type and mutated sequences. Across the 250 synonymous mutations, we found that for 181 of them (72.4%), CodonMPNN correctly predicts higher likelihoods for the more highly expressed codon sequences (the pairs above the horizontal line in Figure 4).

## 5. Discussion and Future Work

To aid structure-based protein engineering via inverse folding methods, we developed CodonMPNN as a drop-in replacement for ProteinMPNN. By directly generating codon sequences instead of amino acid sequences, CodonMPNN directly generates codon sequences that are closer to codon optimality than naively choosing the most frequent codon per amino acid. Further, the user can condition on the host system in which they aim to express the generated codon sequence - information that often is available to experimentalists but has not been used in previous inverse folding approaches. We experimentally verified that CodonMPNN retains ProteinMPNN's performance and assigns higher likelihoods to more highly expressed codon sequences than lower expression codon sequences that encode the same amino acid sequence.

**Future Work.** In many codon optimization tasks, no protein structure is available, and the goal is to obtain an amino acid sequence's most highly expressed codon sequence for a given host system. While CodonMPNN can be a useful drop-in replacement for inverse folding models such as Pro-

teinMPNN, it fails to address this related task. Thus, we aim to fine-tune a protein language model to generate codon sequences conditioned on amino acid sequences and our taxon label.

## 6. Acknowledgements

We thank Sergey Ovchinnikov for his advice and Yehlin Cho, Peter Mikhael, Felix Faltings, Bowen Jing, Tommi Jaakkola, Bonnie Berger, and Eric Alm for insightful discussion.

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

# A. Taxonomy tree partitioning

We use the following algorithm to cluster our proteins according to taxonomic information. We create groupings with $k \in \{50, 100, 500, 1000, 5000, 10000, 20000, 50000\}$ and find that $k = 20000$ leads to optimal performance for the conditioned CodonMPNN.

---

**Algorithm 1** Balanced Tree Partitioning

---

1: **Input:** Tree $tree$, number of clusters $k$

2:

3: **function** AssignToCluster($node$, $cluster\_id$):

4:     $clusters[cluster\_id]$.append($node$.name)

5:     $cluster\_sizes[cluster\_id] \leftarrow cluster\_sizes[cluster\_id] + 1$

6:     **for** $child$ in $node$.children **do**

7:       **if** $child$.is_leaf() or ($cluster\_sizes[cluster\_id]$ + $child$.subtree_size $\leq max\_cluster\_size$) **then**

8:         AssignToCluster($child$, $cluster\_id$)

9:       **else**

10:         $next\_cluster\_id \leftarrow$ index of min($cluster\_sizes$)

11:         AssignToCluster($child$, $next\_cluster\_id$)

12:       **end if**

13:     **end for**

14:

15: $clusters \leftarrow \{\}$

16: $cluster\_sizes \leftarrow [0] * k$

17: $max\_cluster\_size \leftarrow (tree$.subtree_size $+ k - 1)//k$

18: AssignToCluster($tree$, 0)

19: **return** $clusters$

---

