# OpenReview forum: "CodonMPNN for Organism Specific and Codon Optimal Inverse Folding"
_ICML.cc/2024/Workshop/ML4LMS — ML4LMS Poster_

### Official Review · Reviewer_eit2 · 2024-06-03
**CodonMPNN enables protein sequence design at the DNA level, conditioned on structure (proteinMPNN) and taxonomy**

**Rating:** 9
**Confidence:** 5

**Review:**

# Manuscript Review: "CodonMPNN for Organism Specific and Codon Optimal Inverse Folding"

The manuscript "CodonMPNN for Organism Specific and Codon Optimal Inverse Folding" describes an algorithm which includes taxonomy aware codon level design of protein sequence. Leveraging the very successful proteinMPNN structure embeddings and incorporating those with taxonomy embeddings (for further taxon conditioning). This notion of DNA level encoding which here it is combined with structural encoding (protein sequence labels in the case of the proteinMPNN encoder) is important in many cases when native proteins are being experimented in more esoteric cells. The author further incorporated a balanced taxonomy clustering method which is supposed to avoid results bias. This topic is central in current research and development teams both in academia and industry and offers additional degrees of freedom in the design of novel proteins for example for therapeutics purposes.

## Notes:
- The example (only one given in the text) using synonymous mutations for protein belonging to a yeast system is not the case of an esoteric expression system, as the yeast expression system is widely used. It would be interesting to see benchmark the system over a truly esoteric expression system.
- In many real world design pipelines the generation of a functional protein is still the bottleneck, i.e. Lower TM scores for sequences over a prediction oracle. The fair comparison Table 1 shows proteinMPNN is still superior to codonMPNN in that important benchmark. A further discussion about this could be informative and relevant to the way embeddings from both DNA and structure are considered.
- Paragraph 3 of the introduction is a bit misleading where the design pipeline is broken into 3 steps because the first step is not being altered. A fair description would be to focus over design steps 2 and 3 using proteinMPNN for example to a single pass using codonMPNN.
- Because of this claim (paragraph 3) and results of Table 1, it would be interesting to have an additional comparison between codon generated by codonMPNN for “fixed” (proteinMPNN generated) sequences to codonMPNN freely generated DNA sequences.

## Bullet Points:
- **Quality** - The manuscript is well organized and well written. Some references to acknowledge previous fundamental work in the field is in order, e.g., (Rosenberg, A.A., Marx, A. & Bronstein, A.M. "Codon-specific Ramachandran plots show amino acid backbone conformation depends on identity of the translated codon." Nat Commun 13, 2815 (2022). [https://doi.org/10.1038/s41467-022-30390-9](https://doi.org/10.1038/s41467-022-30390-9)).
- **Clarity** - The author clearly presents the topic, data, methods, benchmarks, and results.
- **Originality** - The manuscript introduces an important topic in protein design. This work with further development could allow significant progress to the field of protein design both in academia and industry.
- **Significance** - The manuscripts may have an immediate impact given code is released, less so if not. Code release would be crucial to rapidly benchmark this algorithm compared to many others which are currently being released which uses a similar fundamental MPNN architecture.

## Conclusion:
This work introduces an important study that aims to improve experimental success rate and better our understanding in cell conditioned expression yield. Code availability would contribute to the progress of this study by allowing rapid community benchmarking  over experimental datasets including on more esoteric cell lines to prove the method generality.

---

### Official Review · Reviewer_PkrD · 2024-06-11
**CodonMPNN introduces an inverse folding method for codon design**

**Rating:** 10
**Confidence:** 5

**Review:**

The authors introduce a straightforward modification of the ProteinMPNN architecture, where they design for codons instead of protein sequence. They also add a conditional tag of a host system.

The proposed method is well laid out and appropriate baselines are set: CodonMPNN shows better codon recovery than a randomly initialized sequence, in addition to retraining ProteinMPNN performance. Preliminary experimental validation also verifies their claim about expression.